# Interaction between Mesenchymal Stem Cells and Immune Cells during Bone Injury Repair

**DOI:** 10.3390/ijms241914484

**Published:** 2023-09-23

**Authors:** Wenjing Xu, Yumei Yang, Na Li, Jinlian Hua

**Affiliations:** Shaanxi Centre of Stem Cells Engineering & Technology, College of Veterinary Medicine, Northwest A&F University, Yangling, Xianyang 712100, China; xuwenjing@nwafu.edu.cn (W.X.); 3331364521@nwafu.edu.cn (Y.Y.)

**Keywords:** bone fracture, MSCs, immune cells

## Abstract

Fractures are the most common large organ trauma in humans. The initial inflammatory response promotes bone healing during the initial post-fracture phase, but chronic and persistent inflammation due to infection or other factors does not contribute to the healing process. The precise mechanisms by which immune cells and their cytokines are regulated in bone healing remain unclear. The use of mesenchymal stem cells (MSCs) for cellular therapy of bone injuries is a novel clinical treatment approach. Bone progenitor MSCs not only differentiate into bone, but also interact with the immune system to promote the healing process. We review in vitro and in vivo studies on the role of the immune system and bone marrow MSCs in bone healing and their interactions. A deeper understanding of this paradigm may provide clues to potential therapeutic targets in the healing process, thereby improving the reliability and safety of clinical applications of MSCs to promote bone healing.

## 1. Introduction

The process of bone regeneration is intricate and well-coordinated, with fracture healing being a multi-stage process. The involvement of immune cells in this process is crucial, although their primary activity occurs during the initial stages of fracture healing (Figure 1) [1]. Throughout the different phases of bone healing, cytokines that possess both inflammatory and immune-regulating functions, such as interleukin (IL)-1β [2], IL-6 [3,4], IL-17F [2,5], IL-23 [2], and tumor necrosis factor (TNF) [6,7], play significant roles. Acute inflammation following bone injury promotes fracture healing. B cells infiltrate fracture healing tissue in large numbers shortly after injury and differentiate into plasma cells that secrete a large number of factors, including osteoprotegerin, to promote bone healing by inhibiting osteoclastogenesis [8]. Inflammatory signaling by TNF-α, IL-1, and other pro-inflammatory cytokines has also been shown to exert a pro-regenerative function through injury-induced changes in the tissue microenvironment (such as expression of cell surface receptors) [9]. Activation of the inflammatory response drives nuclear factor–kappa B (NF-κB)-mediated gene expression to further amplify inflammation [10]. However, chronic long-term inflammation has an inhibitory effect on fracture healing [11]. For example, TNF-α and IL-1 are directly associated with bone destruction in patients with arthritis. In mice with inflammatory arthritis, treatment with TNF-α inhibitors enhanced healing, and lack of IL-1 due to genetic mutations prevented bone and joint disease [12]. It has also been shown that the chronic inflammatory environment caused by peri-implantitis recruits a large number of neutrophils and induces degradation of the extracellular matrix, leading to more differentiation of mesenchymal stem cells (MSCs) towards adipocytes than osteoblasts. On the other hand, reactive oxygen species (ROS) are involved in the chemotaxis of MSCs towards adipocytes, and eventually, the osteolysis process is activated due to the imbalance of osteogenic effects [13]. Immune cells of the innate immune system such as neutrophils, macrophages, monocytes, and natural killer cells are activated and recruited to the fracture site together with a variety of cytokines, thus forming an inflammatory response to create conditions for initial fracture healing. Osteoprogenitor mesenchymal stem cells are also involved in this process [14,15].

Immune cells and related cytokines in the acquired immune response also play an important regulatory role in promoting the progress of the healing [16,17]. For example, T helper (Th) 1 cells are polarized by IL-12 and secrete interferon (IFN)-γ, IL-2, and TNF-α, mainly affecting macrophages. For example, TNF-α mediates increased receptor activator of nuclear factor kappa-B ligand (RANKL) expression in macrophages, thereby stimulating osteoclastogenesis [18]. B cells produce approximately 40–60% of the total bone marrow-derived osteoprotegerin (OPG) and thereby inhibit osteoclast differentiation [19]. MSCs participate in both innate and adaptive immunity, and their immunomodulatory functions are exerted mainly via interactions with immune cells through cell-to-cell contact and paracrine activity [20,21,22].

Transplantation of allogeneic bone marrow mesenchymal stem cells in mice has been shown to induce a Th1 type immune response, with significant increases in T cells, B cells, macrophages, and IFN-γ, inhibiting bone formation. This was due to significant inhibition of the expression of osteocalcin, runt-related transcription factor 2 (*Runx2*), and alkaline phosphatase genes in the implants [23]. Liu et al. reported that pro-inflammatory T cells secrete large amounts of IFN-γ and TNF-α and induce apoptosis of MSCs through the combined effect of down-regulating the RUNX2 pathway and enhancing TNF-α signaling [24]. Some scholars have also found that delayed fracture healing is significantly correlated with the enrichment of CD8+ T cells in hematomas [25]. This is consistent with these findings in animal studies.

In 2019, the results of a European human clinical trial showed that surgical treatment using bioceramic particles combined with autologous MSCs for patients with nonunion of tibia, femur, and humerus is safe and feasible [26]. Another clinical study showed that bone marrow (BM) MSCs enriched by the screen–enrich–combine circulating system (SECCS) for surgical treatment promoted bone regeneration in patients with post-fracture nonunion [27]. As of January 2019, there have been more than 700 clinical trials using MSC transplants as an alternative to common therapies for better outcomes [28].

MSCs can promote the migration of autologous BM-MSCs and nerve growth at the fracture site in the treatment of fracture nonunion [29,30] and differentiate to chondrocytes and osteoblasts under certain induction. They can also promote the differentiation of osteoblasts [31], thus promoting bone remodeling [32]. Compared with common therapy, the rate of bone healing is greatly accelerated and the success rate of fracture healing is improved, although MSC treatment of fractures is not 100% cured [33]. Calori, G.M. et al. indicated that the healing rate of fractures treated with MSC was close to 90% [34]. The experiments performed by Chu, W. et al. showed that out of 16 patients who received mesenchymal stem cell/β-tricalcium phosphate composites (MSC/β-TCP), 15 achieved excellent or good recovery within 2 years, and out of 23 patients who received porous β-TCP alone, 14 achieved good recovery. During follow-up, one patient treated with porous β-TCP alone had a Lysholm score of less than 60, indicating poor recovery [35]. The current clinical trial results show that MSCs can be used for fracture healing with a good healing effect and a high cure rate.

In this review, we discuss the current understanding of the interaction between MSCs and the immune system in fracture repair.

## 2. Interaction between MSCs and Macrophages

### 2.1. Effects of Macrophages on MSC

Pro-inflammatory cytokines (TNF-α, IL-6, IL-1β, and IFN-γ) and oncostatin M (OSM) stimulate the osteogenic differentiation of MSCs through OSM and nuclear factor–kappa B (NF-κB) pathways [36,37]. In addition, Cu-mesoporous silica nanosphere (MSN)/macrophage-conditioned medium (CM) upregulates OPG in BM-MSCs and downregulates RANKL to inhibit osteoclast formation [38]. In a co-culture experiment, macrophages phagocytosed by carbon nanohorn-(CNH-) also expressed OSM, which accelerated the osteogenic differentiation of MSCs through the signal transducer and activator of transcription 3 (STAT3) signaling pathway [37]. Lu et al. demonstrated that lipopolysaccharide (LPS)-induced M1 macrophages promote osteogenic formation through the cyclooxygenase-2 (COX2)–prostaglandin E2 (PGE2) pathway [39]. Tu et al. explained the stimulating effect of pro-inflammatory macrophages on MSC osteogenesis from another perspective. IL-23 secreted by macrophages directly induces bone formation of BM-MSCs by activating STAT3 and β-catenin. When the IL-23 p19 antibody neutralized IL-23 in the macrophage CM, the calcium formation and alkaline phosphatase (ALP) activity of MSCs were reduced [40].

However, in the study by Gong et al. M2 macrophages enhanced the osteogenic differentiation of BM-MSCs, while M1 macrophages disrupted their osteogenic differentiation. Pro-regenerative cytokines such as transforming growth factor (TGF)-β, vascular endothelial growth factor (VEGF), and insulin-like growth factor 1 (IGF-1) are produced by M2 macrophages, and harmful inflammatory cytokines such as IL-6, IL-12, and TNF-α are produced by M1 macrophages. This is suspected to be a mechanism that regulates osteogenic differentiation [41]. Meanwhile, TGF-β and VEGF have been shown to have angiogenic activity [42].

### 2.2. Effects of MSCs on Polarization of Macrophages

MSCs also affect macrophage subtypes. The implanted MSCs generated a cascade of events, including the recruitment of BM-derived specific progenitors with vasculogenic and osteogenic properties, resulting in the mobilization of cells of the innate immune system such as macrophages and the induction of their functional switch from a pro-inflammatory to an anti-inflammatory proresolving phenotype. MSCs induce M1 to M2 macrophages, which is attributed to the activation of the NF-κB pathway by PGE2 secreted by exogenous MSCs [43]. Li et al. also found a transformation from M1 to M2 in their study of bone defects with the bone-inducing material laponite (LAP). Although LAP contributes to bone regeneration, it is still related to inflammation as a foreign body. They found that MSCs transformed LAP-induced M1 macrophages into M2 phenotypes, creating an anti-inflammatory/pro-lytic environment that promotes bone formation [44]. However, the MSCs were not detected 4 weeks after transplantation, suggesting that early MSCs may regulate the polarization of macrophages [45,46]. Related literature reports that macrophages are transformed into an M2 type through cytokine-pretreated MSC and IL-4 secreting MSC [47,48]. Although pre-treated BM-MSCs and IL-4-secreting BM-MSCs can promote bone formation, the time of bone regeneration in vitro has a significant impact. After co-culture with macrophages, pretreated MSC promoted bone regeneration in the early stage (day 3), while IL-4 secreting MSC played a role in the later stage (day 7). Based on the secretion of IL-4 and PGE2, IL-4-secreting MSCs also have a greater immunomodulatory ability in the transformation from M1 to M2 [49,50]. The interaction between MSC and macrophages is shown in Figure 2.

## 3. MSCs Inhibit Dendritic Cells (DCs)

As an antigen-presenting cell (APC), DCs phagocytose and processes antigens before presenting them to T lymphocytes via MHC molecules on the cell surface to activate an acquired immune response [51]. Differentiation of monocytes into DCs secretes IL-12, which can help I CD4+ T cells differentiate into Th1 cells. It has been shown that allogeneic BM-MSCs inhibit the ability of monocytes or CD34+ hematopoietic progenitors to differentiate into DCs, as well as their ability to secrete pro-inflammatory cytokines [52]. Meanwhile, allogeneic BM-MSC can promote the release of anti-inflammatory factors and inhibit the polarization of initial CD4+ lymphocytes into pro-inflammatory cells [53,54]. Similar to T lymphocytes, allogeneic MSCs may also inhibit DC function by means of cell–cell contact [55]. This process, in which MSCs secrete TGF-β1, simultaneously decreases the expression of DC co-stimulatory molecules such as CD80 and CD86 [56,57]. One study found that when autologous or homologous MSCs were co-cultured with dendritic cells, dendritic cells induced differentiation of primitive T cells into classical Treg [56]. Another study found that when monocyte-derived DCs were co-cultured with allogeneic MSCs, the DCs were more likely to adopt myeloid-derived suppressor cell (MDSC)-like phenotypes in response to the growth-regulating oncogene γ (GRO-γ) chemokines secreted by the MSCs. Dendritic cells co-cultured with autologous or homozygous MSCs can also induce initial T-cell differentiation to classical Treg [56]. Homozygous MSCs co-cultured with monocyte-derived dendritic cells secrete growth-regulating oncogene γ (GRO-γ) chemokines, which promotes the development of DCs into a myeloid-derived suppressor cell (MDSC)-like phenotype. GRO-γ-treated MDSCs had a tolerogenic phenotype that was characterized by an increase in the secretion of IL-10 and IL-4 and a reduction in the production of IL-12 and IFN-γ [58]. Recent studies have shown that MSCs inhibit the maturation of DCs by upregulating suppressor of cytokine signaling 1 (*socs1*) through IL-6-mediated cytokine signal transduction, thereby blocking the TLR1 signaling pathway [59].

## 4. Interaction between MSCs and Natural Killer (NK) Cells

NK cells belong to the core cells of the natural immune system. NK cells, unlike all other white blood cells, are capable of recognizing and attacking foreign cells, cancer cells, and viruses on their own. On the one hand, secretion of inflammatory cytokines secreted by NK cells can be inhibited by MSCs, such as IFN-γ or TNF-α. On the other hand, intercellular factors secreted by MSCs can inhibit NK cell activity [60]. TGF-β and IL-6 produced by activated MSCs limit NK cell effector function but promote NK cell differentiation [61]. In addition, experimental data have suggested that MSCs can improve the function of NK cells after severe injury [62].

NK cells are able to be activated not only by recognizing cells lacking major histocompatibility complex (MHC) Class I surface molecules, but also by cytokines including IL-2 and IL-12 [63]. Activated NK cells induce apoptosis mainly by releasing cytotoxic particles including perforin, NK cytotoxic factor, and TNF [63]. Allogeneic BM-MSCs sharply inhibit IL-2-induced resting NK cell proliferation while having little effect on activated NK cell proliferation [64]. The same study demonstrated that IL-2-activated NK cells initiate NK cell-mediated MSCs cytotoxicity by binding to specific ligands on MSCs, which effectively degrades endogenous and exogenous MSCs. However, IFN-γ can inhibit this cleavage by using upregulated human leukocyte antigen (HLA) class I molecules [64,65]. A recent study has shown that allogeneic MSCS can adapt their immune behavior in the inflammatory environment by activating toll-like receptor 3 (TLR3) to resist the killing effect of IL-2-activated NK cells [66].

## 5. Interaction between MSCs and Neutrophils

In the process of healing fractures, MSCs have been found to exert inhibitory effects on various immune cells, with the exception of neutrophils [67]. MSCs inhibit neutrophil apoptosis, alter their chemotaxis, and enhance their respiratory burst capacity [68,69,70]. When activated, MSCs work synergistically by producing IL-6, IFN-γ, and granulocyte-macrophage colony-stimulating factor (GM-CSF) to effectively delay neutrophil death [71]. Furthermore, MSCs secrete IL-8 and macrophage migration inhibitory factor (MIF) to attract neutrophils and facilitate their infiltration into sites of inflammation. As a result of this recruitment process, the recruited neutrophils exhibit heightened antimicrobial activity and an increased respiratory burst capacity [69,72]. In a corneal injury model and a peritonitis mouse model, TNF alpha-induced protein 6 (TSG-6) secreted by MSCs effectively inhibits the entry of neutrophils into the injury site, thereby significantly reducing inflammatory responses [73,74]. MSCs also suppress neutrophils through multiple pathways to decrease immune responses and reduce tissue damage [73,75]. Such a biphasic effect indicates the plasticity of immune regulation in MSCs, and we speculate whether MSCs can trigger the necessary inflammatory response while suppressing excessive immune responses and maintaining a local microenvironment conducive to tissue repair.

## 6. Interactions between MSCs and Mast Cells (MCs)

The functions of MCs in bone repair mainly modulate angiogenesis as well as anabolic and catabolic processes during fracture repair and remodeling [76]. It is speculated that MSCs can induce an appropriate inflammatory response while suppressing excessive immune reactions. This implies that during the healing process, MCs may also impact the behavior of MSCs by interacting with specific receptors such as IL-1 receptor (IL-1R), IL-6R, TNF receptor (TNFR), CXCR1, TGF-β receptor 1 (TGFβRI), or the basic FGF receptor (bFGFR) [77]. Conversely, MSCs release factors like TGF-β, VEGF, or IL-6, which can influence MC function through their respective receptors (TGFβR1/2, VEGF receptor, IL-6R) [78,79]. Meanwhile, mast cell tryptase is an additional angiogenesis factor. Mast cells act at the sites of new vessel formation by secreting tryptase, which then functions as a potent angiogenesis factor [42]. Additionally, it is possible that MSCs have a direct impact on MC behavior during the healing of fractures by regulating factors such as MC numbers, cytokine production, mediator release, and degranulation. As a result, the influence of MCs on MSCs in bone healing may be attributed to their ability to affect the migration, proliferation and differentiation of MSCs. In summary, while recent studies have identified specific roles for MCs in fracture healing, further investigation is needed to fully understand how these cells interact with MSCs in this context [80].

## 7. MSCs Inhibit B Lymphocytes

Μmt gene knockout of B lymphocytes may lead to changes in the immune environment of the new bone induction site, thereby stimulating the initial accumulation and proliferation of mesenchymal progenitor cells [81]. As an important part of the acquired immune response system, B cells can differentiate into plasma cells under the stimulation of antigen, which can synthesize and secrete antibodies (immunoglobulin), mainly performing humoral immunity of the body. Early studies have shown that mouse allogeneic BM-mscs have an inhibitory effect on the proliferation, activation, and immunoglobulin G (IgG) secretion of B cells [82]. The presence of allogeneic BM-MSCs can inhibit the proliferation of by arresting B-lymphocytes in the G (0)/G (1) phase of the cell cycle. Meanwhile, MSCs can also inhibit the proliferation of B lymphocytes by producing soluble factors in transwell experiments [83,84]. Allogeneic BM-MSCs can also change the activation mode of extracellular response kinase 1/2 and the p38 mitogen-activated protein kinase pathway, both of which are involved in the survival, activation, and proliferation of B cells [84]. Data from another study suggest that humoral factors released by BM-MCS inhibit terminal differentiation of B cells, possibly by inhibiting the expression of mature protein-1 in B lymphocytes [85]. The inhibition of B cell activation by allogeneic MSCs seems to depend on IFN-γ and cell contact through programmed cell death-1/programmed cell death–ligand 1 (PD-1/PD-L1) interactions, like the immunosuppression of T lymphocytes by MSC [86]. However, this is not consistent with data showing that MSCs promote B cell proliferation and differentiation in vitro [87].

## 8. MSCs Inhibit T Lymphocytes

In the anabolic and catabolic stages of fracture healing, both innate and adaptive immune processes are essential. In addition to macrophages and neutrophils clearing the damaged site, specific cell-mediated immune functions also clear necrotic tissue during the initial inflammatory stage after injury, promote angiogenesis, and initiate repair [88].

Tregs are known to have a positive effect on fracture healing because they inhibit the secretion of pro-inflammatory cytokines by activated T cells, which is conducive to the differentiation of MSCs into osteoblasts [89]. Through direct cell–cell contact [90] or paracrine [91], MSCs play a dual role in regulating these different T-cell subpopulations during bone healing. On the one hand, BM-MSCs can inhibit the activation and proliferation of T cells (CD4^+^ and CD8^+^ subpopulations) by inducing cell cycle G0 arrest, as shown in the in vitro mixed lymphocyte reaction [92,93,94]. In addition, MSCs stimulate T cell apoptosis through Fas/FasL-dependent pathways [95] and PD-L1 secretion [96]. On the other hand, MSC can not only induce the formation of CD4^+^CD25^+^Foxp3^+^Tregs [97,98] but can also secrete heme oxygenase-1 (HO-1) and IL-10 and upregulate PD-1 receptors on Tregs, triggering the immunosuppressive ability of Tregs [99].

Interestingly, bone fractures cause the immune system to be suppressed [100] and a local increase in the number of induced T regulatory (iTREG) cells that inhibit the active adaptive immune response in the fracture callus [2]. Research has indicated that MSCs are capable of maintaining a state of low immunogenicity [101,102] by producing paracrine factors with immunosuppressive properties [103,104,105] through their direct impact on immune cell populations, such as T cells [106,107]. Such effects indicate that these cells transmit immune tolerance during the early stages of endochondral bone formation and protect developing tissues by inhibiting the allogeneic proliferation of T cells during stem cell recruitment and cartilage formation [2].

## 9. Effects of B Lymphocytes and T Lymphocytes on MSCs

B cell-mediated regulation functions in the early stages of the bone healing process, and its absence is associated with delayed healing or non-healing in fractures. Upregulation of Breg function may also contribute to improved fracture healing [108]. B cells may also inhibit bone formation by suppressing osteoblast differentiation. For example, in patients with rheumatoid arthritis, B cells have been reported to inhibit osteogenesis via TNF-α and C-C motif chemokine ligand (CCL) 3 [109]. Similarly, the maturation of osteoblasts is hindered by B cells in the presence of colony-stimulating factor 3 (G-CSF) during the process of hematopoietic stem cell and progenitor cell homing [80]. MSC-mediated bone regeneration can be partially impeded by CD8+ T cells, while osteogenic differentiation is completely inhibited by CD4+ T cells. However, Treg infusion can eliminate the inhibition of MSCs by activated T cells [24,80]. Pro-inflammatory T cells inhibit bone formation by secreting cytokines that inhibit the RUNX-2 pathway and promote apoptosis of bone marrow mesenchymal stem cells [24]. Meanwhile, activated T cells induce MSC cell apoptosis via the Fas/FasL and CD40/CD40L pathways [67]. However, contrary to the view that T cells inhibit bone healing, some studies have shown that certain types of T cells are beneficial to osteogenesis. For example, Treg cells can resist this negative effect because they are able to shed large amounts of TNF receptor superfamily member 1B (TNFRII) and thus inhibit the action of TNF [110], promoting bone formation. IL-17A secreted by γδT cells shifts the lipid differentiation capacity of MSCs to osteogenic differentiation, thereby improving the quality of bone healing [111]. In addition, TNF-α-stimulated T cells also activate the NF-κB pathway and secrete more CCL 5 to recruit MSCs to the site of injury, thus achieving ectopic osteogenesis [112].

## 10. Effects of pH

Bone injury in the form of fracture or osteotomy can lead to vascular disruption. Vascular disruption following bone injury can lead to an acidic, hypoxic wound environment. This disruption of blood flow results in a hypoxic zone accompanied by local tissue ischemia, decreased pH, and elevated lactate levels within the fracture-healed tissue [113]. Previous research has indicated that the pH level decreases from a normal physiological value of 7.4 to 6.8 during the initial two days of healing, which is known as the inflammatory phase [114]. During this phase, mesenchymal stem cells (MSCs) migrate to the injury site to assist in tissue repair. In view of this, Hazehara-Kunitomo, Y. et al. conducted a study on how brief exposure to acidic conditions (pH 6.8) affects the characteristics of BM-MSCs. Interestingly, their findings revealed that pre-treating BM-MSCs with an acidic pH enhances the expression of specific markers associated with stem cell properties (organic cation/carnitine transporter4 (OCT-4) and Nanog homeobox (NANOG) and promotes cell viability and proliferation. However, it was observed that acidic conditions have a negative impact on BM-MSC migration ability. These results suggest that maintaining an acidic pH during the early stages of bone healing is crucial for enhancing BM-MSCs stemness and function [114]. It has been shown that hypoxic cultured hBM-MSCs displayed higher levels of pro-survival, pro-proliferation, pro-migration, vasculogenesis, and angiogenesis genes [115]. However, hypoxia inhibits osteogenic differentiation of MSCs [116]. Therefore, acidic, hypoxic conditions may improve the intrinsic properties of these cells, and these findings may further aid in the development of materials and methods for more precise and in situ control of BM-MSC stemness and function. This is essential for the development of successful cellular therapies.

## 11. Effects of Exosomes on Fractures

Exosomes are metastatic “paracrine factors” that act as signaling molecules in the tissue microenvironment [117]. MSC-derived exosomes are extracellular nano-vesicular products derived from MSCs [118]. Exosomes derived from MSC improve wound healing through signaling or by influencing the fate decisions of certain immune cells, thereby promoting the restoration of immune homeostasis, or through the attenuation of excessive inflammation [119]. Recently, it has been shown that exosome-rich mineral-doped poly-L-propyleneglycolate acidic scaffolds can improve the osteogenesis of human adipose-derived MSCs [120]. There has also been research suggesting that exosomes derived from BM-MSCs alleviate bone loss by restoring the function of recipient BM-MSCs and activating *Wnt/β-catenin* signaling [121]. Hypoxia preconditioning promotes fracture healing by activating hypoxia inducible factor 1, alpha subunit (HIF-1α) to mediate the increase of exosome microRNA 126 (mir-126) [122].

## 12. The Effect of Immune Cells on Angiogenesis

Blood vessels in bone tissue play crucial roles in the repair of injuries. The involvement of vascular endothelial cells (Ecs) in fracture repairs is facilitated through their interaction with bone cells [123]. In bone tissues, two subtypes of vascular Ecs have been identified: type H vascular Ecs characterized by a high expression of CD31 and EMCN, and type L vascular Ecs characterized by a low expression of CD31 and EMCN [124]. Although type H Ecs constitute only 1.77% of all bone Ecs and 0.015% of total bone marrow Ecs, they are surrounded by a significant number of bone progenitor cells that are capable of differentiating into osteoblasts and osteocytes. On the other hand, type L vessels have minimal surrounding bone progenitor cells, suggesting that type H vessels may strongly promote the regeneration process [124,125]. Extensive research has demonstrated that inflammatory responses and neovascularization are critical factors initiating bone regeneration [126,127,128]. The stimulator of interferon genes (STING), an essential factor involved in innate immunity, has been found to be closely associated with angiogenesis. A study indicated that activation of STING leads to a decrease in H-type blood vessels and almost no callus mineralization; however, inhibiting mmSTING can enhance the formation of H-type blood vessels. This suggests that inhibiting STING can accelerate the healing process while promoting the formation of H-type blood vessels simultaneously [129]. Gao et al.’s findings revealed that macrophages/monocytes can differentiate into tartrate-resistant acid phosphatase (TRAP) mononuclear cells on periosteal surfaces during bone modeling. These TRAP mononuclear cells secrete platelet-derived growth factor type BB (PDGF-BB), which induces periostin expression from periosteum-derived cells (PDC). Consequently, PDC recruitment occurs on periosteal surfaces to support both type H vessel formation and osteogenesis processes [130].

In response to tissue damage, the immune system triggers a protective inflammatory reaction following a fracture. This reaction is initiated by the release of various dangerous associated molecular patterns (DAMP) and recognized by pattern recognition receptors (PRRs) on bone marrow Ecs [131]. As a result, complex signaling pathways are activated, leading to the secretion of proinflammatory cytokines like interleukin, TNF, and IFN-α [132]. Moreover, Ecs increase the production of granulocyte colony-stimulating factor (G-CSF) and granulocyte macrophage colony-stimulating factor (GM-CSF), which promote proliferation, migration, and differentiation of hematopoietic stem cells (HSCs) [133]. Simultaneously, hypoxic regions develop within the bone marrow due to inflammation. In response to this hypoxia, Ecs and HSCs upregulate VEGF/VEGFR2 expression to enhance angiogenesis [132]. Since angiogenesis is initiated in response to tissue hypoxia, this leads to elevated expression of EC-derived hypoxia-inducible factor 1-alpha (HIF-1α). This in turn induces upregulation of other proangiogenic factors. Additionally, proteases are released during the inflammatory process that contribute to degradation and remodeling of the extracellular matrix (ECM). Consequently, vasodilation occurs in the affected area along with basement membrane degradation, pericyte detachment, EC migration, and enhanced vascular permeability [134,135].

An increase in regulatory T helper cells and the expression of anti-inflammatory cytokine IL-10 were observed alongside an increase in angiogenic factors (HIF1a and HIF1a regulated genes), specifically within regenerative bone hematoma rather than soft tissue hematoma. These findings suggest that timely resolution of inflammation and early initiation of revascularization are mutually dependent and crucial for promoting a regenerative healing process [128]. Recent discoveries have found that lymphatic vessels exist in bone, and lymphatic vessels play a key role in promoting bone repair through the IL-6 signaling pathway. Lymphangiogenesis can be used as a therapeutic approach to stimulate hematopoiesis and bone regeneration [136].

## 13. Effect of Aging on Fracture Healing

The incidence of fractures significantly increases with age and presents greater challenges for older patients [137]. As individuals age, the number of osteoblasts decreases. This results in a decline in bone quality and impaired fracture healing due to reduced bone generation processes. A noticeable decline in osteoprogenitor cells was observed in aged mice’s long bones, which correlated with a significant decrease in H-type vessels [124]. Molecular changes specific to age were detected in the endothelium of tissues, leading to vessel loss and influencing pericyte transformation into fibroblasts. Additional research has indicated that vascular depletion due to aging serves as an early indication of cellular senescence. Consequently, the degree and progression of vascular depletion may determine the limitations on tissue regeneration. Meanwhile, pericytes undergo age-dependent differentiation into fibroblasts. Finally, vascular loss drives cellular changes. These findings imply that strategies to inhibit age-dependent changes in the vasculature, such as loss of vascular abundance and pericyte-to-fibroblast differentiation, have the potential to delay or even prevent cellular dysfunction during aging [138].

In addition, systemic low-level chronic inflammation is present in elderly fracture patients, which can also impair the initiation of bone regeneration in elderly patients. The prolonged inflammatory phase is attributed to continuous high expression of inflammatory cytokines. Additionally, aged macrophages demonstrate heightened sensitivity and responsiveness to inflammatory signals while experiencing decreased proliferation. Failure to transition macrophages from the M1 phenotype to the M2 phenotype results in persistent chronic inflammation, leading to increased activation of osteoclasts and reduced formation of osteoblasts during the healing process. Consequently, this imbalance leads to elevated bone resorption and diminished bone formation during healing. Furthermore, aging contributes to a decline in both the quantity and proliferative capacity of MSCs, further impairing bone healing capabilities. MSCs are more prone to senescence and possess lower potential for generating new bone tissue. As a result, age-related changes in interactions between macrophages, osteoclasts, and MSCs contribute significantly towards compromised bone healing abilities [138].

## 14. Application of Single-Cell Analysis in Fracture Repair

While the roles of immune cells and stromal cells in fracture healing have been suggested, there is still limited understanding regarding the variations in their numbers or types during different stages of fracture healing. Zhang et al. utilized single-cell RNA-seq (scRNA-seq) to examine immune cells in a mouse fracture model and discovered significant differences in B-cell populations, with fewer B cells observed during the tissue formation stage and higher numbers during the tissue healing stage. These findings indicate the crucial involvement of B cells in promoting fracture healing. Furthermore, exosomes released by B cells were found to hinder osteoblast differentiation and enhance osteoclast formation, thereby reducing osteogenic activity [139]. Avin et al. using scRNA-seq analysis, compared the proportions of immune cell subtypes and identified higher levels of monocytes and CD14 DCs as well as lower levels of T cells, myelocytes, and promyelocytes within the nonunion bone group. This study also provided valuable insights into gene expression changes from an osteoimmunological perspective. Such discoveries contribute significantly towards comprehending the mechanisms underlying bone nonunion [140]. In order to gain insights into the regulatory factors governing the differentiation of bone mesenchymal cells and their lineage specialization, Sivaraj et al. employed scRNA-seq to investigate the functional characteristics, lineage differentiation potential, and cellular fate transitions of bone mesenchymal stromal cells. Their findings revealed that diaphysis mesenchymal stromal cells, which exhibited limited differentiation capacity, were associated with the sinusoidal vasculature within the bone microenvironment. On the other hand, metaphysis mesenchymal stromal cells displayed multilineage potential for osteogenic, adipogenic, and chondrogenic differentiation. Moreover, PDGFRb signaling and Jun-B transcription factor were identified as key regulators controlling the fate of bone mesenchymal stromal cells during bone formation processes by influencing cell proliferation and differentiation [141]. To enhance fracture healing treatments effectively, future investigations should employ a combination of scRNA-seq with complementary techniques to explore the crucial mechanisms underlying immune cell dynamics and variations in stromal cell behavior throughout all stages of bone repair. Additionally, exploring the interactions between stromal cells and immune cells during fracture healing may offer valuable insights for identifying novel therapeutic targets.

## 15. Main Limitations and Prospects

The potential of MSCs in promoting bone repair has been promising, but there are still several factors that may limit their use and increase variability between studies. These include the source of MSCs, as well as the timing and quantity of implantation. Additionally, it is important to fully characterize the population of MSCs and determine the best approach based on the type and location of bone trauma or defects [142]. In cases where in vitro expansion is necessary for cell transplantation or gene delivery purposes, limitations arise due to a lack of standardization and automation in this process. Variability can occur depending on factors such as the number of passages, media type, serum usage (including lot-to-lot variations), animal-free media availability, or the presence of xenogenic proteins, which pose a risk for contamination. Similar to other genetically modified cells, genetic modulation presents safety concerns, particularly when viral vectors are used [143]. Intrinsic biological variability between tissue sources, donors, clonal subsets, single-cell variability, and extrinsically introduced variability through non-standard isolation, selection, and production methods result in MSC heterogeneity. Translation of MSC products beyond clinical trials has been limited due to, among other factors, MSC heterogeneity, complicating the ability to produce a reliable, uniform therapy throughout the MSC expansion and banking process [22]. Nevertheless, with advancements in molecular biology techniques, there is an increasing possibility to utilize engineered MSCs or their secretomes in clinical settings for addressing bone defects.

The choice of biomaterials also affects treatment outcomes. The body’s immune response to bone implants is influenced by a number of factors, including the icroporosity of the biomaterial, the surface microstructure, the material hardness, and the particle size [144,145]. Macrophages are the main effector cells of the immune response to implants, and bone implants affect MSCs mainly in terms of cell viability, adhesion, migration, proliferation, and differentiation [146]. The presence of surface roughness in titanium, such as polished, machined, and grit-blasted commercially pure titanium, has been observed to impact the attachment and spreading of immune cells. Over time, macrophage adhesion increases on all surfaces, while cell spreading is enhanced with higher surface roughness levels [147]. Furthermore, the roughness of titanium can also influence the production of inflammatory cytokines and chemokines by macrophages. Significant stimulatory effects have been noted on sandblasted and acid-etched surfaces [148], whereas the modified surfaces of titania nanotube arrays exhibit reduced immune responses compared to raw titanium surfaces [149]. It is worth noting that a bone’s surface roughness measures approximately 32 nm at a nanoscale level, making nanomaterials highly biomimetic [150]. In vitro studies have demonstrated that nanoscale microstructures effectively stimulate human MSCs to produce bone minerals, even without osteogenic supplements [151]. Additionally, it has been discovered that macrophages exposed to a microstructured topography become activated with both M1 and M2 characteristics instead of nanostructured topography alone [152]. Research also indicates that periosteal extracellular matrix (PEM) hydrogels promote the recruitment and M2 polarization of macrophages. They further facilitate differentiation of MSCs into endothelial-like cells, along with human umbilical cord vascular endothelial cell (HUVEC) tube formation and osteogenic differentiation of MSCs [153]. Therefore, new strategies that reduce MSC heterogeneity through pooled-MSCs, clonal MSCs, priming pretreatment of MSCs, and biomaterial-MSC interactions to improve their immunomodulatory potencies will clinically impactful [22,154].

In the clinic, because bone marrow-derived MSCs are more difficult to collect and are themselves less abundant, studies of alternative sources have emerged [155,156,157]. Some research results show that adipose MSCs and BM-MSCs have similar immunophenotypes and in vitro differentiation abilities, as well as similar immunomodulatory abilities. They also show that adipose MSCs are more metabolically active, secrete more cytokines, and are more efficient in their immunomodulatory effects than BM-MSCs. Adipose-derived stem cells (ASCs) can be considered as a good alternative to BM-MSCs for immunomodulatory therapy [158]. ASCs are a significant subset of MSCs [159]. ASCs, derived from the stromal vascular fraction (SVF), are progenitor cells located around blood vessels [160]. Similar to BM-MSCs, ASCs have been extensively studied in the field of bone tissue engineering [161]. ASCs possess the ability to differentiate into multiple cell lineages, including adipocytes, chondrocytes, and osteoblasts. The fate determination of these lineages is regulated by key factors such as Runx2 and Osterix (Sp7 transcription factor) [125,162]. Various signaling pathways play a crucial role in controlling osteogenic differentiation, including bone morphogenetic protein (BMP) [163], Notch [164], Wnt [165], and Hedgehog signaling [166]. Among these pathways, the Wnt signaling pathway acts as a pivotal regulator that directs ASC differentiation towards osteogenesis by upregulating *Runx2* and *Osterix* expression levels [167]. Numerous studies have explored different substrates to enhance the osteogenic potential of ASCs. These substrates include vitamin D3 [168], alendronate [169], selenium [170], platelet-rich plasma [171], and the inflammatory response [172,173,174,175].

In conclusion, the mechanisms of the immunomodulatory effects of MSCs are complex, and characterizing a single factor is difficult because of the interaction between various factors (Figure 3). More in-depth studies are needed to fully understand and exploit the mechanisms of immunomodulation by MSCs.

## 16. Conclusions

Convincing evidence supports the important role of BM-MSCs in the bone healing process, which is mainly due to their immunomodulatory ability through cell-to-cell contact and by secreting paracrine factors. Clinical trials have verified their relative safety and effectiveness, as well as their potential use in cell therapy for bone diseases with potentially inflammatory conditions. However, despite the great progress made since the discovery and characterization of MSCs, it is necessary to evaluate the osteogenic potential of MSCs by selecting donors and/or subpopulations with high osteogenic abilities to promote bone recovery and regulate inflammation. Further research will clarify the mechanisms of bone marrow MSC immune regulation to promote bone repair.

## Figures and Tables

**Figure 1 ijms-24-14484-f001:**
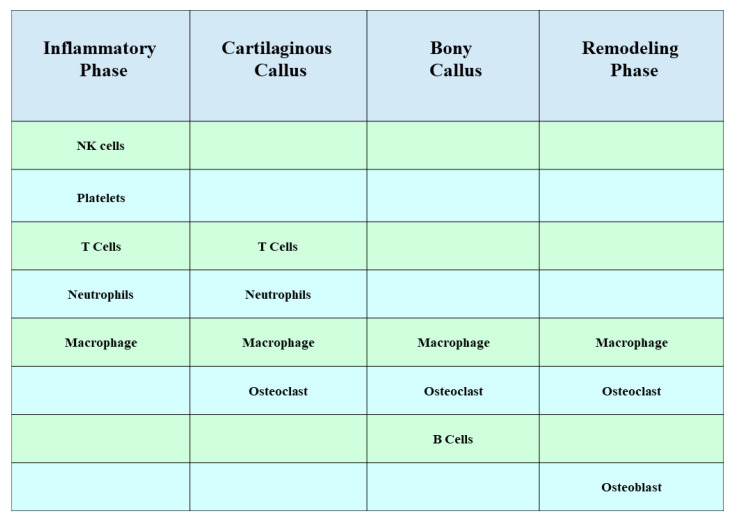
Role of immune cells during the four stages of fracture healing. Most of the activity of immune cells occurs early in fracture healing, and macrophages accompany almost the entire process of fracture healing.

**Figure 2 ijms-24-14484-f002:**
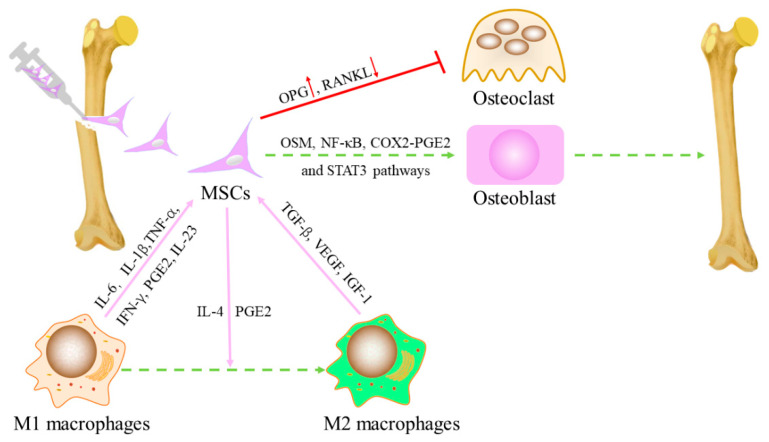
The interaction between MSCs and macrophages. Macrophages regulate the recruitment and differentiation of MSCs. On the upper part of the image, the best-known macrophage pro-inflammatory cytokines are shown, as well as the osteoinductive factors involved in the regulation of MSC functions. MSCs reciprocally regulate macrophage recruitment and function, usually with PGE2 and IL-4-mediated transformation of macrophages from M1 type to M2 type. MSCs up-regulate OPG and down-regulate RANKL to inhibit osteoclast formation.

**Figure 3 ijms-24-14484-f003:**
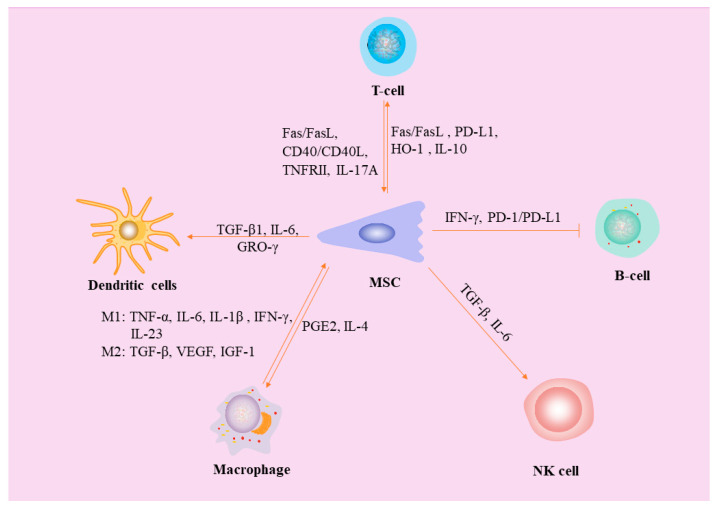
The mechanism of interaction between MSC and immune cells. Pro-inflammatory cytokines secreted by macrophages (TNF-α, IL-6, IL-1β, IL-23, and IFN-γ) stimulate osteogenic differentiation of MSCs, and TGF-β and VEGF have been shown to have angiogenic activities. PGE2 and IL-4 secreted by MSCs induce the transformation of macrophages from M1 type to M2 type. MSCs affect DC differentiation and proinflammatory cytokine secretion through TGF-β1, IL-6, and GRO-γ. MSCs not only stimulate T cell apoptosis through Fas/FasL pathway and PD-L1 secretion, but can also secrete heme oxygenase-1 (HO-1) and IL-10, upregulate PD-1 receptors on Tregs, and trigger the immunosuppressive ability of Tregs. Meanwhile, activated T cells induced MSCs apoptosis via the Fas/FasL and CD40/CD40L pathways. However, Tregs are able to shed large amounts of TNF receptor superfamily member 1B (TNFRII) and thus inhibit the action of TNF and promote bone formation. IL-17A secreted by γδT cells shifts the lipid differentiation capacity of MSCs to osteogenic differentiation, thereby improving the quality of bone healing. TGF-β and IL-6 produced by activated MSCs limit NK cell effector functions but promote NK cell differentiation. The inhibition of B cell activation by MSC depends on IFN-γ and PD-1/PD-L1.

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
