# Peer review of "Interaction between Mesenchymal Stem Cells and Immune Cells during Bone Injury Repair"

_ijms, 2023, doi:10.3390/ijms241914484_

Round 1
Reviewer 1 Report (New Reviewer)
1/ In general, this manuscript would benefit from reorganization to first describe the host immune response, including the key cell types and cytokines involved in the use of mesenchymal stem cells (MSCs) for cellular therapy of bones injury. This should be followed by discussion of clinical trials that have been developed and a description of the specific cells and cytokines that were induced and correlated with reduced the effect of bone pathology in specific clinical or animal models.
2/ Please be sure to define all acronyms when they are first used in the manuscript.
3/ Figure 1 could be reorganized and presented in a better way.
4/ The title should reflect the context of the review correctly.
Author Response
Please see the attachment.

Reviewer 2 Report (New Reviewer)
This is a very intresting review. Fractures are a prevalent form of major organ trauma in humans. Initially, the inflammatory response supports bone healing in the early stages post-fracture, but when inflammation becomes chronic or persistent due to factors like infection, it hinders the healing process. The precise mechanisms governing the regulation of immune cells and their cytokines in bone healing are not yet fully understood. An emerging clinical treatment approach involves using mesenchymal stem cells (MSCs) for bone injuries. These bone progenitor MSCs not only have the ability to differentiate into bone but also interact with the immune system, thereby promoting the healing process. This review examines both in vitro and in vivo studies concerning the immune system's role and the interactions of bone marrow MSCs in bone healing. A deeper understanding of this interplay may offer insights into potential therapeutic targets, enhancing the reliability and safety of clinical applications of MSCs to facilitate bone healing.
The topic is timely. The review is indeed well-written and timely. However, it's crucial to acknowledge a significant omission.
The review should have included the critical aspect of vascular involvement and immune crosstalk during bone repair PMID: 34281770. Notably, recent discoveries have revealed the presence of lymphatic vessels in bones, which play a pivotal role in promoting bone repair through IL6 signaling. PMID: 36669473 This addition would have further enriched the discussion and deepened our understanding of the topic.
Additionally, the impact of aging on mesenchymal and perivascular cells, as well as cytokines, has not been addressed PMID: 33536212. Since fracture healing poses a significant challenge in the elderly population, it is essential to include and thoroughly discuss this aspect in the review.
Minor edits required
Round 2
Reviewer 2 Report (New Reviewer)
Authors have addressed all my comments and I have no further comments
This manuscript is a resubmission of an earlier submission. The following is a list of the peer review reports and author responses from that submission.
Round 1
Reviewer 1 Report
The authors made an attempt to review the reparation potential of MSCs in the inflammatory milieu. However, the paper is rather messy and represents a lot of repeats, contradictions, conglomerations of facts, which require profound analysis and systematization. The language is hard to understand.
Other comments
1. Lines 12-14; 47-49; 40-41;116-117;143-146;194-195; 200-201206-207;252-253; 294-295; 306-307;335 (effects of exosomes on what?) These sentences are not clear. They should be clarified.
2. Line 42: Reference of Chu is required.
3. Lines 61-62 vs 68-69; 63-64vs72-73; 106-108 vs269-270; 95-96 vs 275-276: All the repeated fragments should be removed.
4. Fig 2 is not clear. The authors should describe what they wished to present.
5. Lines 215-218 represent contradictory statements. This should be corrected.
6. No reference and explanations for Fig3 in the text. This should be corrected.
The authors made an attempt to review the reparation potential of MSCs in the inflammatory milieu. However, the paper is rather messy and represents a lot of repeats, contradictions, conglomerations of facts, which require profound analysis and systematization. The language is hard to understand.
Other comments
1. Lines 12-14; 47-49; 40-41;116-117;143-146;194-195; 200-201206-207;252-253; 294-295; 306-307;335 (effects of exosomes on what?) These sentences are not clear. They should be clarified.
2. Line 42: Reference of Chu is required.
3. Lines 61-62 vs 68-69; 63-64vs72-73; 106-108 vs269-270; 95-96 vs 275-276: All the repeated fragments should be removed.
4. Fig 2 is not clear. The authors should describe what they wished to present.
5. Lines 215-218 represent contradictory statements. This should be corrected.
6. No reference and explanations for Fig3 in the text. This should be corrected.
Reviewer 2 Report
The authors of this review have provided insight in the immune response and bone marrow MSCs for fracture repair. The emphasis of the manuscript is on the osteogenesis of MSCs and their interaction with NK cells, macrophages and dendritic cells. This is a nice review and shows the importance of immune cells in fracture repair.
The following points need to be addressed in their review,
1. Angiogenesis is a critical process in fracture repair and involves complex interaction with immune cells (e.g. macrophages) and osteoprogenitors. Apart from the section on pH, there is a lack of information on these relationships in previous sections. Authors should include more information on immune cells and angiogenic cells within the context of fracture repair in the previous sections.
2. Figure 2 appears an incomplete figure and authors should create a more appropriate figure.
3. Macrophage sections show the importance of the surrounding environment of polarization of the cells, specifically IL-4 for M2 macrophages and IFN-gamma for M1 macrophages. Apart from the description of laponite, how do specific materials (e.g. titanium) and their surface architecture influence the immune response ? May bone implants elicit a response in MSCs and thus, the interaction with immune cells. This needs to be acknowledged in this review.
4. Are there any differences in the immune cell interaction between adipose and bone marrow MSCs ? Is there literature that has sought to understand whether there are different interactions between these MSC sources
5. Has RNAseq or single cell RNAseq undercover specific mechanisms that control the response in immune cells relative to MSC presence during bone healing ? Authors should include more mechanistic insights in their review.
The English language quality is appropriate.
Round 2
Reviewer 2 Report
The authors have addressed my comments appropriately.